# RAMP1 Signaling Mitigates Acute Lung Injury by Distinctively Regulating Alveolar and Monocyte-Derived Macrophages

**DOI:** 10.3390/ijms251810107

**Published:** 2024-09-20

**Authors:** Atsushi Yamashita, Yoshiya Ito, Mayuko Osada, Hiromi Matsuda, Kanako Hosono, Kazutake Tsujikawa, Hirotsugu Okamoto, Hideki Amano

**Affiliations:** 1Department of Molecular Pharmacology, Graduate School of Medical Sciences, Kitasato University, Sagamihara 252-0373, Japan; ayamashi@med.kitasato-u.ac.jp (A.Y.); yito@kitasato-u.ac.jp (Y.I.); hosono@med.kitasato-u.ac.jp (K.H.); 2Department of Anesthesiology, Kitasato University School of Medicine, Sagamihara 252-0374, Japanokasuke@med.kitasato-u.ac.jp (H.O.); 3Department of Pharmacology, Kitasato University School of Medicine, Sagamihara 252-0374, Japan; 4Department of Emergency and Critical Care Medicine, Kitasato University School of Medicine, Sagamihara 252-0374, Japan; 5Laboratory of Molecular and Cellular Physiology, Graduate School of Pharmaceutical Sciences, Osaka University, Osaka 565-0871, Japan; tujikawa@phs.osaka-u.ac.jp

**Keywords:** RAMP1, macrophage, neutrophil, nerve, immunity

## Abstract

Acute respiratory distress syndrome (ARDS) is a life-threatening lung injury that induces cytokine hypersecretion. Receptor activity-modifying protein (RAMP) 1, a subunit of the calcitonin gene-related peptide (CGRP) receptor, regulates the production of cytokines. This study examined the role of RAMP1 signaling during lipopolysaccharide (LPS)-induced acute lung injury (ALI). LPS administration to wild-type (WT) mice depleted alveolar macrophages (AMs) and recruited monocyte-derived macrophages (MDMs) and neutrophils. RAMP1-deficient (RAMP1^−/−^) mice exhibited higher lung injury scores, cytokine levels, and cytokine-producing neutrophil infiltration. RAMP1-deficient AMs produced more cytokines in response to LPS than WT AMs. Adoptive transfer of RAMP1-deficient AMs to RAMP1^−/−^ mice increased cytokine levels and neutrophil accumulation compared to the transfer of WT AMs. RAMP1^−/−^ mice had reduced MDM recruitment and lower pro-inflammatory and reparative macrophage profiles. Cultured bone marrow (BM)-derived RAMP1-deficient macrophages stimulated with LPS showed decreased expression of pro-inflammatory and pro-repairing genes. CGRP administration to WT mice reduced cytokine production and neutrophil accumulation. These findings indicate that RAMP1 signaling mitigates LPS-induced ALI by inactivating AMs and promoting inflammatory and repair activities of MDMs. Targeting RAMP1 signaling presents a potential therapeutic approach for the treatment of ARDS.

## 1. Introduction

Acute respiratory distress syndrome (ARDS) is a life-threatening lung injury with a mortality rate of up to 40% [1]. It is characterized by damage to the alveolar endothelial cells and inflammatory cell infiltration. Vascular damage causes vascular permeability and edema infiltration, leading to tissue hypoxia. Marked elevation of inflammatory cytokine levels, often referred to as a “cytokine storm”, plays a crucial role in the progression of ARDS, along with the accumulation of immune cells, including macrophages and neutrophils [2].

Lung tissues are innervated by peripheral sensory nerves originating from vagal and spinal sensory neurons. Calcitonin gene-related peptide (CGRP), released from sensory nerve endings, innervates the lungs [3,4]. CGRP^+^ nerve fibers are extensively distributed along the airways [5]. The action of CGRP is mediated by the CGRP receptor, which comprises two subunits: calcitonin receptor-like receptor (CLR) and receptor activity-modifying protein 1 (RAMP1). RAMP1 regulates the specific binding of CGRP to CLR [6,7]. RAMP1 serves as a rate-limiting factor for the transport of the receptor to the plasma membrane, thereby regulating its trafficking [8]. In addition to its vasoactive properties, RAMP1 also regulates immunity [9]. We previously showed that CGRP is responsible for the development of inflammation in the liver [10] and intestines [11] by attenuating pro-inflammatory cytokine production through RAMP1 signaling in macrophages. In the lungs, CGRP expression participates in protecting against pathogens and regulating immune reactions [12]. In a rat model of ARDS established through intraperitoneal administration of lipopolysaccharide (LPS), pretreatment with CGRP significantly decreased protein influx and capillary leakage in the early phase of acute lung injury [13]. Treatment with the CGRP receptor antagonist CGRP8-37 aggravated the lung injury [14]. In hospitalized patients with COVID-19, low serum CGRP levels have been observed, suggesting that normal CGRP levels serve a protective and therapeutic role [15]. In contrast, high plasma CGRP levels in critical patients with COVID-19 were associated with hyperinflammation, as evidenced by increased cytokine levels, including interleukin (IL)-6 [16]. However, the role of RAMP1 signaling in ARDS remains unclear.

Pulmonary macrophages play a crucial role in maintaining pulmonary function and protecting against respiratory pathogens [17]. Alveolar macrophages (AMs), which reside in the alveolar airspace, are the first line of defense against pathogens in the lungs while maintaining proper lung function [18]. During respiratory infections, monocyte-derived macrophages (MDMs) originating from the bone marrow (BM) [19,20,21] are recruited to the lung tissues to support antimicrobial responses and combat infections [22]. Although both AMs and MDMs contribute to the pulmonary immune response by secreting cytokines and chemokines, it remains unclear whether RAMP1 signaling regulates cytokine production by affecting macrophage activity during the progression of ARDS.

In this study, we examined the role of RAMP1 signaling in LPS-induced acute lung injury (ALI) and elucidated the regulatory mechanisms by which RAMP1 signaling participates in the development of ARDS. To investigate the role of RAMP1 signaling, we utilized a mouse model of ARDS induced by the administration of LPS, a component of the outer membrane of gram-negative bacteria, rather than a virus. Although the accumulation of various immune cell types in the lungs involved in the promotion of ARDS differs between LPS-induced ALI and virus-induced ARDS, both establish a cytokine storm, a characteristic feature of ARDS.

## 2. Results

### 2.1. RAMP1 Signaling Deficiency Increases Mortality, Lung Injury, and Pulmonary Edema in LPS-Induced Acute Lung Injury

We examined the role of RAMP1 signaling in LPS-induced ALI based on mortality rates monitored every 24 h up to 168 h after LPS administration. Mortality was defined by comprehensive pathological examinations during necropsy, which revealed macroscopically observable discoloration and hemorrhage of pulmonary tissue, as well as clinical manifestations including respiratory alterations (characterized by superficial respiration and elevated spontaneous respiratory rate), hypoactivity, weight reduction, and hypothermia. After LPS administration, all wild-type (WT) mice survived, while the survival rates in RAMP1^−/−^ mice dropped at 48 h, and further decreased to 60% at 96 h (Figure 1A). We compared the two groups from 0–72 h and found that RAMP1^−/−^ mice had a higher lung injury score at 72 h after LPS administration than WT mice (Figure 1B). The total protein (TP) concentration in the bronchoalveolar lavage fluid (BALF) of RAMP1^−/−^ mice was higher than that from WT mice (Figure 1C). These results indicated that RAMP1^−/−^ mice were susceptible to LPS-induced ALI, suggesting that inhibition of RAMP1 signaling aggravated LPS-induced ALI and pulmonary edema.

### 2.2. Expression of CGRP/RAMP1 during Acute Lung Injury

We determined the expression of CGRP and RAMP1 during LPS-induced ALI. The concentrations of CGRP in the BALF of WT and RAMP1^−/−^ mice increased at 72 h after LPS administration (Figure 1D). Considering that the DRG is a source of CGRP, we assessed CGRP expression in thoracic Th_4-12_ DRGs by immunostaining. CGRP immunoreactivity was enhanced in DRGs from both WT and RAMP1^−/−^ mice (Figure 1E). The number of CGRP^+^ cells in the DRGs from WT and RAMP1^−/−^ mice at 72 h was greater than that at 0 h, but there was no difference in CGRP^+^ cell levels between the groups. The pulmonary mRNA levels of *Cgrp* in WT mice increased at 6 h and decreased thereafter (Figure 1F); although *Cgrp* mRNA levels were reduced at 72 h, those in WT mice were higher than those in RAMP1^−/−^ mice. We also observed CGRP expression in lung tissues by immunohistochemical analysis. CGRP^+^ nerve fibers are distributed along the airways, and, in some areas, CGRP^+^ cells terminate in pulmonary neuroendocrine cells (PNECs) at 72 h [23,24].

The mRNA levels of *Ramp1* in WT mice increased at 72 h to higher levels than those in RAMP1^−/−^ mice (Figure 1G). Immunofluorescence analysis of the lung tissues demonstrated that RAMP1 was co-expressed in CD68^+^ cells at 0 h and 72 h. RAMP1 was partially co-expressed in myeloperoxidase (MPO)^+^ cells at 72 h. These results indicated that RAMP1 originates from macrophages and neutrophils during LPS-induced ALI.

### 2.3. RAMP1 Signaling Deficiency Increases Cytokine Production

Considering that RAMP1^−/−^ mice exhibited aggravation of LPS-induced ALI, we examined whether RAMP1 signaling regulated proinflammatory cytokine production. After LPS administration, both WT and RAMP1^−/−^ mice showed elevated levels of pro-inflammatory cytokines, including tumor necrotic factor alpha (TNF-α), IL-1β, and IL-6, in the BALF (Figure 2A). The levels of these cytokines were higher in RAMP1^−/−^ mice than in WT mice. Similarly, RAMP1^−/−^ mice had higher mRNA levels of *Tnfa*, *Il-1b*, and *Il-6* at 6 h and 24 h than WT mice (Figure 2B). These results indicated that RAMP1 deficiency enhanced pro-inflammatory cytokine production during LPS-induced ALI.

### 2.4. RAMP1 Signaling Deficiency Enhances Neutrophil Accumulation and Attenuates MDM Accumulation

We analyzed the LPS-induced accumulation of inflammatory cells in the inflamed lungs (Figure 3A, Appendix A). At basal levels, the most abundant CD45^+^ cells were AMs (~95%)—resident pulmonary macrophages—in both WT and RAMP1^−/−^ mice (Figure 3B). At 72 h, both WT and RAMP1^−/−^ mice showed a significant increase in CD45^+^ cells; the number of CD45^+^ cells in RAMP1^−/−^ mice was greater than that in WT mice. Similarly, neutrophils accumulated extensively in the lungs over time after LPS administration. RAMP1^−/−^ mice displayed greater neutrophil recruitment at 72 h than WT mice. LPS administration reduced AMs in both groups at 6 h and remained low at 24 and 72 h. Instead, the levels of MDMs increased at 72 h after LPS administration [20]; the number of MDMs in RAMP1^−/−^ mice was lower than that in WT mice. Considering that neutrophils outnumbered all other immune cells at 72 h, the accumulation of neutrophils in LPS-treated lung tissues appeared to be critical for LPS-induced ALI.

Considering that immune cells were recruited to lung tissues after LPS administration, we determined the chemokines responsible for driving immune cell recruitment (Figure 3C). The protein and gene levels of chemokine C-X-C-Motif ligand 2 (CXCL2), a neutrophil chemokine, and C-C motif chemokine ligand-2 (CCL2) were higher in RAMP1^−/−^ mice than in WT mice (Figure 3D).

### 2.5. Roles of AMs in RAMP1 Signaling during ALI

AMs are a source of pro-inflammatory cytokines during ALI, indicating that AMs contribute to the initiation of LPS-induced ALI. To examine the role of AMs in LPS-induced ALI, AMs were removed by intratracheal (i.t.) administration of clodronate liposomes (CL) or control liposomes (Cont) 24 h before LPS administration. Flow cytometry revealed that CL treatment reduced AM levels in WT and RAMP1^−/−^ mice at 6 h and 72 h (Figure 4A). In contrast, CL treatment increased the number of CD45^+^ cells, including neutrophils, in WT mice at 72 h compared with that in Cont-treated WT mice. CL treatment did not change the number of CD45^+^ cells and neutrophils in RAMP1^−/−^ mice at 72 h compared with that in Cont-treated RAMP1^−/−^ mice. There was no difference in neutrophil accumulation between CL-treated WT and RAMP1^−/−^ mice, suggesting that neutrophil recruitment was induced by RAMP1 signaling in AMs. MDM levels remained unchanged within 6 h after LPS administration. At 72 h, CL-treated RAMP1^−/−^ mice showed higher MDM levels than Cont-treated RAMP1^−/−^ mice. Although MDMs in WT mice tended to increase compared with those in Cont-treated WT mice, the difference was not significant. These results suggested that AMs inhibit the excessive recruitment of neutrophils and MDMs in lung tissues exposed to LPS.

Treatment of WT mice with CL was also accompanied by an increased lung injury score (Figure 4B). The lung injury score in CL-treated WT mice was higher than that in Cont-treated WT and RAMP1^−/−^ mice. The total protein (TP) concentration in the BALF at 72 h in CL-treated WT and RAMP1^−/−^ mice was higher than that in Cont-treated WT and RAMP1^−/−^ mice. RAMP1^−/−^ mice treated with CL and Cont displayed higher TP levels. Alternatively, LPS-induced ALI was enhanced in both groups after removal of AMs, suggesting that the pulmonary-protective activity of RAMP1 signaling is at least partly derived from AMs.

We determined the levels of pro-inflammatory cytokines and chemokines in the BALF (Figure 4C). At 6 h, CL treatment reduced the concentrations of TNF-α and IL-1β but not IL-6, compared with the Cont treatment in WT and RAMP1^−/−^ mice. These results suggested that AMs contributed to the production of pro-inflammatory cytokines (TNF-α and IL-1β) during the initiation of LPS-induced ALI. In contrast, at 72 h, the CL treatment increased the levels of TNF-α and IL-1β but not IL-6, compared with the Cont treatment in WT and RAMP1^−/−^ mice. These results suggested that AMs participated in the regulation of pro-inflammatory cytokines (TNF-α and IL-1β) during the inflammatory phase of LPS-induced ALI and that cells other than AMs produced abundant pro-inflammatory cytokines. Considering that CL treatment induced extensive neutrophil accumulation in the lungs, we determined CXCL2 levels at 72 h. The levels of CXCL2 were higher in the BALF of CL-treated mice than in Cont-treated mice, indicating that increased CXCL2 levels induced the overaccumulation of neutrophils in the lungs of CL-treated WT and RAMP1^−/−^ mice (Figure 4D). The levels of CCL2, a chemoattractant for macrophages, in both CL-treated groups were higher than those in the Cont-treated groups, suggesting that increased CCL2 levels promoted the accumulation of MDMs in CL-treated mice. Collectively, these results indicated that AMs participated in the production of cytokines and chemokines to promote LPS-induced ALI.

We examined whether RAMP1 signaling regulates cytokine production in isolated AMs in vitro. When AMs isolated from WT and RAMP1^−/−^ mice were stimulated with LPS, the protein levels of TNF-α, IL-1β, and IL-6 were higher in AMs from RAMP1^−/−^ than those in WT mice (Figure 5). Additionally, the administration of CGRP reduced the levels of TNF-α and IL-6 in WT AMs but not RAMP1^−/−^ AMs, indicating that the CGRP/RAMP1 signaling pathway displays an inhibitory effect on cytokine production from AMs. These results suggested that exacerbated LPS-induced lung injury in RAMP1^−/−^ mice may be caused by the excessive production of cytokines and chemokines by AMs during the initiation phase of ALI. Therefore, AMs initiated LPS-induced ALI, and RAMP1 signaling in AMs suppressed the excessive production of cytokines and chemokines to induce the accumulation of neutrophils in lung tissues. Alternatively, RAMP1 signaling in AMs may protect mice from LPS-induced ALI.

To further establish the protective function of RAMP1 signaling in AMs, WT or RAMP1-deficient AMs were adoptively transferred to RAMP1^−/−^ mice (Figure 6). The numbers of CD45^+^, neutrophils, and MDMs, but not AMs, were lower following transfer with WT-derived AMs than those with RAMP1^−/−^-derived AMs (Figure 6A). RAMP1^−/−^ mice treated with WT-derived AMs showed reduced lung injury scores, TP concentrations in the BALF, and IL-1β levels (Figure 6B–D). Although the other cytokines, including TNF-α and IL-6, tended to have higher levels in RAMP1^−/−^ mice treated with RAMP1^−/−^-derived AMs than those with WT-derived AMs, there were no significant differences. These results suggested that RAMP1 signaling in AMs mitigated LPS-induced lung injury by reducing cytokine production and inflammatory cell recruitment.

### 2.6. MDMs in RAMP1 Signaling Contribute to LPS-Induced ALI

AM levels decreased during the progression of LPS-induced ALI, whereas MDM levels increased in inflamed lung tissues (Figure 3). At 72 h, MDM levels in inflamed lung tissues from WT mice were higher than those in RAMP1^−/−^ mice. We investigated the involvement of MDMs in LPS-induced acute lung injury based on the inflammatory mediators in BALF-isolated MDMs at 72 h. The mRNA expression of pro-inflammatory cytokines, including *Tnfa*, *Il-1b*, and *Il-6*, in MDMs from RAMP1^−/−^ mice was lower than that in WT mice (Figure 7A). Flow cytometry revealed that MDMs at 72 h were positive for IL-1β, and the percentage of IL-1β^+^-MDMs was higher in WT mice than in RAMP1^−/−^ mice (Figure 7B). However, we observed lower percentages of TNFα^+^ and IL-6^+^ MDMs than IL-1β^+^ MDMs. Considering that MDMs are derived from BM [20], we determined the levels of cytokines in cultured BM-derived macrophages. Stimulation of BM macrophages with LPS increased the protein levels of TNF-α, IL-1β, and IL-6 in WT mice in a RAMP1 signaling-dependent manner, suggesting that RAMP1 signaling in BM-derived macrophages increases cytokine levels in response to LPS (Figure 7C). These results indicated that MDMs from WT mice showed a pro-inflammatory phenotype at 72 h.

In contrast, MDMs from WT mice also had higher granulocyte macrophage colony stimulating factor (GM-CSF) and Il-10 mRNA levels than those from RAMP1^−/−^ mice (Figure 7D). Although GM-CSF exerts pro-inflammatory actions, including macrophage and neutrophil recruitment at inflammatory sites, it is an indispensable factor for AM differentiation, survival, and homeostasis, suggesting that GM-CSF has important reparative effects in lung tissues [25]. The increases in mRNA levels of Gm-csf and Il-10 in cultured BM-derived macrophages of WT mice stimulated with LPS were dependent on RAMP1 signaling (Figure 7E). These results indicated that MDMs from WT mice also showed a pro-reparative phenotype. Collectively, MDMs from WT mice exhibited both a pro-inflammatory and reparative phenotype at 72 h.

### 2.7. RAMP1 Signaling Suppresses Accumulation of Neutrophils

Substantial neutrophil infiltration was evident in the lung tissues of RAMP1^−/−^ mice at 72 h (Figure 3). Although pro-inflammatory cytokine levels were markedly elevated at 6 h (the initiation phase of LPS-induced ALI), the protein levels of TNF-α, IL-1β, and IL-6 in RAMP1^−/−^ mice remained high at 72 h (the injury phase), indicating that neutrophil accumulation was accompanied by increased inflammatory cytokine levels (Figure 2). Therefore, we examined whether neutrophils release these cytokines, thereby contributing to severe ALI. Accordingly, we determined the percentage of TNF-α^+^, IL-1β^+^, and IL-6^+^ neutrophils at 72 h. The percentages of TNF-α^+^ and IL-1β^+^ neutrophils were greater than those of IL-6^+^ neutrophils, which were found to be few (Figure 8A). The percentages of TNF-α^+^ and IL-1β^+^ neutrophils were higher in RAMP1^−/−^ mice than in WT mice. However, there was no significant difference in the percentage of IL-6^+^ neutrophils between the two groups. The *Tnfa* and *Il-1b* mRNA levels in neutrophils from RAMP1^−/−^ mice were greater than those in WT mice (Figure 8B). The mRNA levels of *Ramp1* were upregulated in neutrophils as well as MDMs, which was consistent with the immunofluorescence analysis results (Figure 1G). These findings suggested that infiltrating neutrophils contribute to the progression of LPS-induced ALI through the release of pro-inflammatory cytokines and that RAMP1 signaling in neutrophils suppressed overproduction of pro-inflammatory cytokines.

Considering that IL-6 was not generated from the accumulated neutrophils, we sought other sources of IL-6 during LPS-induced acute lung injury. A possible source of IL-6 appeared to be MDMs; however, the number of IL-6^+^ MDMs was insubstantial (Figure 7B). Another possible source is epithelial cells. Immunofluorescence revealed that IL-6 expression was co-localized with aquaporin 5 (AQP5)- and prosurfactant protein C (SFC)-positive cells, suggesting that Type I and Type II cells produced IL-6 in response to LPS treatment (Appendix A).

To further understand the involvement of lung neutrophils in ALI, mice were treated with an anti-Ly6G antibody to inhibit neutrophil activity at 0 h and 24 h after LPS treatment. Compared with WT mice treated with control IgG, WT mice treated with anti-Ly6G Ab showed reduced TP concentrations in the BALF and inflammatory cytokine levels, including TNF-α, IL-1β, and IL-6, at 72 h (Appendix A). Treatment with anti-Ly6G Ab reduced the number of neutrophils but did not affect the number of AMs (Appendix A). The number of MDMs in anti-Ly6G Ab-treated WT mice was higher than that in control IgG-treated WT mice. These results suggested that infiltrating neutrophils contribute to LPS-induced acute lung injury by releasing cytokines.

### 2.8. CGRP Treatment Mitigates LPS-Induced ALI

Finally, we examined whether activation of RAMP1 by CGRP treatment mitigated LPS-induced ALI. At 72 h, CGRP-treated WT mice displayed reduced infiltration of pulmonary cells, including CD45^+^, neutrophils, and AMs, but not MDMs (Figure 9). Compared to the vehicle, treatment with CGRP reduced TP concentrations in the BALF along with the lung injury score, which were associated with decreased mRNA levels of pro-inflammatory cytokines (*Il-1b* and *Il-6*) and chemokines (*Cxcl2* and *Ccl2*).

## 3. Discussion

Our study showed that the deletion of RAMP1 signaling exacerbated ALI, increased cytokine and chemokine production, and promoted cytokine-producing neutrophil accumulation. The inhibition of AM activity increased cytokine levels during the inflammatory phase and enhanced lung injury and neutrophil accumulation. Isolated RAMP1-deficient AMs produced more cytokines in response to LPS than isolated WT AMs. Adoptive transfer of RAMP1-deficient AMs to RAMP1-deficient mice upregulated cytokine levels and neutrophil accumulation compared to the transfer of WT AMs. RAMP1-deficient mice exhibited reduced recruitment of MDMs, which also showed lower levels of proinflammatory and anti-inflammatory profiles than WT mice. In vitro, RAMP1-deficient BM-derived macrophages stimulated with LPS displayed lower levels of pro- and anti-inflammatory genes relative to WT macrophages. CGRP treatment suppressed lung injury, neutrophil recruitment, and cytokine production. These results indicate that deletion of RAMP1 signaling aggravated ALI by activating AMs and suppressing the pro-inflammatory and repairing functions of MDMs.

In the peripheral lungs, bronchi/bronchioles, vasculature/lymphatics, and alveoli are innervated by substantial amounts of sensory nerves [3,26] and PNECs [24]. Our immunofluorescence analysis revealed that CGRP^+^ nerve fibers extended through the airway and terminated in bronchioles and PNECs, suggesting that these nerves can sense extrinsic stressors in the alveoli, including toxins and pathogens. Increases in CGRP+ neurons have been observed in response to airway inflammation [27,28]. Following LPS administration, the release of CGRP into the alveolar space correlated with increased CGRP^+^ neurons in the thoracic DRGs (Figure 1). The expression of RAMP1, a subunit of the CGRP receptor, was observed in macrophages [7], including AMs and MDMs, and neutrophils [29] during ALI. Although both WT and RAMP1-deficient mice had increased CGRP levels, RAMP1-deficient mice exhibited exacerbated lung injury, suggesting that RAMP1 signaling induces a protective response to mitigate the injury [30]. During LPS-induced ALI, AMs were depleted and MDMs were recruited to lung tissues while neutrophils infiltrated extensively, indicating that RAMP1 signaling participated in the regulation of RAMP1-expressing immune cell activity in response to i.t. LPS administration.

In healthy states, AMs are the most abundant macrophages in lung tissue [31]. AMs maintain tissue homeostasis and act as the first line of host defense against external pathogens. During ALI, AMs orchestrate the initiation and resolution of inflammation to restore homeostasis. CL-treated AM-depleted mice displayed higher mortality rates, pro-inflammatory cytokine (TNF-α and IL-1β) levels and inflammation, and neutrophil recruitment compared to control mice under bacterial exposure [32,33]. Similarly, we found that AM-inhibited WT mice exhibited enhanced lung injury scores, pro-inflammatory cytokine and chemokine production, and neutrophil accumulation at 72 h after LPS administration. These results suggested that AMs protected the lung by inhibiting the endotoxin-induced cytokine and chemokine generation and neutrophil accumulation.

Our study showed that inhibition of AM activity diminished the differences in lung injury, pro-inflammatory cytokine levels, and neutrophil recruitment between WT and RAMP1-deficient mice, suggesting that AMs mediate the protective properties of RAMP1. Consistent with this view, transplantation of RAMP1-deficient AMs aggravated LPS-induced lung injury, cytokine production, and neutrophil accumulation compared with transplantation of WT AMs. Interestingly, we found that inhibition of AM activity with CL reduced the release of pro-inflammatory cytokines, including TNF-α and IL-1β, at 6 h but ultimately exacerbated LPS-induced lung injury at 72 h. In addition, cultured AMs isolated from RAMP1-deficient mice produced higher levels of pro-inflammatory cytokines in response to LPS than those isolated from WT mice. These results indicated that AMs produced pro-inflammatory cytokines during the early stage of lung injury and that the loss of RAMP1 signaling in AMs induced the overproduction of inflammatory cytokines, thereby aggravating LPS-induced lung injury. Collectively, these findings suggested that RAMP1 signaling in AMs prevented overt inflammatory responses to LPS, including the excessive production of pro-inflammatory cytokines and chemokines and the accumulation of neutrophils, thereby mitigating lung injury.

During the initial phase of LPS-induced ALI, AM populations exhibit a significant reduction, which is consistent with previous findings [34]. Upon activation by LPS stimulation, AMs undergo cell death, resulting in decreased numbers of AMs. As dying cells release pro-inflammatory mediators, AM death plays a crucial role in the progression of inflammation during LPS-induced ALI. The reduction in AM populations may also be attributed to their detachment from the epithelial cells. In addition to apoptosis and necrosis, several modes of programmed cell death, including pyroptosis and ferroptosis, have been proposed; however, the precise mechanisms of AM death remain unclear [35]. The repopulation of AMs occurs during the recovery phase of LPS-induced ALI, presumably through their self-renewal and local proliferation [20,34,35].

We demonstrated that RAMP1-deficient mice had fewer MDMs than WT mice but experienced more severe lung injury. To understand the contribution of MDMs to ALI, we determined the cytokine levels in MDMs. Compared to RAMP1-deficient MDMs, RAMP1-expressing MDMs exhibited pro-inflammatory profiles, as evidenced by the increased mRNA expression of *Tnfa*, *Il-1b*, and *Il-6*, consistent with recent results [20,34,36], suggesting that genes related to the pro-inflammatory macrophage phenotype are highly expressed in MDMs at peak inflammation. Similarly, we showed that cultured BM-derived macrophages stimulated with LPS displayed a pro-inflammatory macrophage phenotype in a RAMP1 signaling-dependent manner. On the other hand, MDMs expressed genes related to the anti-inflamamtory phenotype, including *Il-10*, and upregulated expression of *Gm-csf*. Although GM-CSF promotes inflammation by recruiting macrophages and granulocytes to the inflammatory site, it also participates in AM differentiation, survival, and maintenance of homeostasis [25]. Signals from the local microenvironment are important for maintaining the identity and functionality of AMs, and the higher expression of GM-CSF in MDMs from WT mice may contribute to the restoration of AMs and reparation of damaged tissues. Alternatively, the lower expression of GM-CSF in MDMs from RAMP1-deficient mice may delay the recovery of AMs during injury. Further studies are needed to elucidate the contribution of RAMP1 signaling to GM-CSF expression and the role of GM-CSF in MDMs during ALI.

RAMP1-deficient mice with depleted AMs or adoptively transferred RAMP1-deficient AMs displayed aggravated lung injury accompanied by increased MDM levels. Despite the artificial nature of chemical and cell transfer approaches, these results indicated that RAMP1 signaling in MDMs protected against LPS-induced lung injury in the presence of RAMP1-expressing AMs. Collectively, RAMP1-expressing MDMs exhibited both pro-inflammatory and reparative macrophage phenotypes, suggesting that MDMs were responsible for initiating and resolving inflammation in tissue repair [34,37,38].

Our in vitro experiments demonstrated that CGRP treatment reduced cytokine production from AMs and BM-derived macrophages of WT mice, suggesting that RAMP1 signaling attenuated cytokine production in response to LPS. Consistent with this observation, RAMP1-deficient AMs exhibited elevated cytokine levels in response to LPS. However, RAMP1-deficient BM-derived macrophages displayed decreased cytokine production. The disparate responses of RAMP1-deficient macrophages to LPS stimulation may be attributed to the differential expression of LPS receptors, including CD14, between cultured AMs and BM-derived macrophages.

Neutrophils play a critical role in the pathology of ARDS [2]. ARDS is characterized by disruption of the alveolar-capillary barrier, leading to fluid leakage and neutrophil infiltration into the airspace. The current study indicated that LPS-induced ALI resulted in extensive neutrophil recruitment in a RAMP1 signaling-dependent manner. In contrast, CGRP treatment reduced neutrophil recruitment to the lungs, consistent with recent results [39]. In the early phase of septic peritonitis, RAMP1-deficient mice displayed increased neutrophil levels [40]. Persistent inflammation due to neutrophil accumulation is associated with a poor prognosis in ARDS [41]. Our findings indicated that pulmonary neutrophil accumulation was mediated by CXCL2. The accumulated neutrophils secreted inflammatory mediators, including TNF-α and IL-1β, to induce lung injury, consistent with previous findings [42]. In contrast, neutrophil depletion improved LPS-induced ALI, corresponding to reduced levels of total protein, pro-inflammatory cytokines, and chemokines. These results suggested that extensive neutrophil accumulation is critical for the progression of LPS-induced ALI [43].

## 4. Materials and Methods

### 4.1. Animals

Male RAMP1-deficient (RAMP1^−/−^) mice (8 weeks old) were generated as previously described [9]. Male C57BL/6 WT mice (8 weeks old) were purchased from CLEA Japan (Tokyo, Japan). All mice were housed under controlled humidity (50 ± 5%) and temperature (25 ± 1 °C) with a 12 h light/dark cycle and ad libitum access to water and food. The experimental protocols were approved by the Institutional Animal Care and Use Committee of the Kitasato University School of Medicine (approval no. 2023-061). All experimental procedures were conducted in accordance with the guidelines of the Science Council of Japan for animal experiments.

### 4.2. Experimental Protocols

After fasting overnight, mice were anesthetized by intraperitoneal (i.p.) injection with mixed anesthetic agents containing 4.0 mg/kg midazolam (Sandoz, a Novartis division, Basel, Switzerland), 0.75 mg/kg medetomidine hydrochloride (Nippon Zenyaku Kogyo, Fukushima, Japan), and 5.0 mg/kg butorphanol (Meiji Seika Pharma, Tokyo, Japan). LPS (*Escherichia coli* O111:B4, 100 μg; Sigma-Aldrich, St. Louis, MO, USA) in 50 μL phosphate-buffered saline (PBS) was intratracheally (i.t.) administered to mice using a 24 G intravenous (i.v.) catheter (B. Braun Medical, Bethlehem, PA, USA). Following the administration of LPS, the effects of medetomidine were reversed by an i.p. injection of 0.75 mg/kg atipamezole (Nippon Zenyaku Kogyo). In separate experiments, some animals received a single subcutaneous injection of CGRP (5.0 μg/mouse; Peptide Institute, Osaka, Japan) [10] dissolved in 0.2 mL saline immediately, 24 h, and 48 h after LPS administration. To evaluate the natural progression of LPS-induced ALI, the mice were not subjected to therapeutic interventions (mechanical ventilation) throughout the course of this study.

### 4.3. Bronchoalveolar Lavage (BAL)

At the indicated time, the mice were anesthetized with a midazolam-medetomidine-butorphanol mixture. The trachea was exposed and cannulated using a 20 G i.v. catheter (B. Braun Medical) and perfused using sterile cold PBS containing 2 mM EDTA (three times, 1 mL each) using a 1 mL syringe to obtain bronchoalveolar lavage fluid (BALF). After centrifugation (400× *g*, 15 min, 4 °C), BALF supernatants were stored at −80 °C for cytokine measurements. After collecting the BAL, the right lungs were collected for RNA extraction, and animals were euthanized via cervical dislocation.

### 4.4. Immune Cell Depletion

For neutrophil depletion, mice were injected i.p. with 100 μg anti-Ly6G (clone 1A8, in 200 μL volume, Bio X Cell, Lebanon, NH, USA) at 0 h and 24 h after LPS administration. Control mice received 100 μg rat IgG (Bio X Cell). To deplete alveolar macrophages, 50 μL clodronate liposomes (CLs; FormuMax Scientific, Sunnyvale, CA, USA) were delivered intratracheally 24 h before LPS administration [12]. Control mice received an equal volume of control liposomes (FormuMax Scientific).

### 4.5. Flow Cytometry Analysis

After centrifugation of the BALF, the cells were treated with red blood cell lysis buffer (BioLegend, San Diego, CA, USA) for 2 min. BAL cells were resuspended in FACS buffer (BD Biosciences, Franklin Lakes, NJ, USA) and incubated with an anti-mouse CD16/32 antibody (BioLegend) for 10 min to prevent non-specific binding. The cells were stained with a combination of Brilliant Violet 421-conjugated anti-CD45 (30-F11, BioLegend), APC-conjugated anti-Ly6G (1A8, BioLegend), PE/Cy7-conjugated anti-CD11b (M1/70, BioLegend), Brilliant Violet 510-conjugated anti-Ly6C (HK1.4, BioLegend), APC/Cy7-conjugated anti-CD11c (N418, BioLegend), and FITC-conjugated anti-SiglecF (S17007L, BioLegend). Cells positive for 7-aminoactinomycin D (BioLegend) were excluded from the analysis. Samples were acquired using a FACS Verse Cytometer (BD Biosciences). Data were analyzed using Kaluza software v2.1 (Beckman Coulter, Brea, CA, USA) and presented as the number of cells per mL.

Intracellular staining was performed using fixation buffer and Perm/Wash buffer (BioLegend). The treated cells were stained with PE-conjugated anti-TNF-α mAb (MP6-XT22, BioLegend), PE-conjugated anti-IL-1β mAb (NJTEN3, Thermo Fisher Scientific, Waltham, MA, USA), PE-conjugated anti-IL-6 mAb (MP5-20F3, BioLegend), and control immunoglobulin. The cells were analyzed by flow cytometry (FACS Verse Cytometer, BD Biosciences). Data were analyzed using Kaluza software v2.1 (Beckman Coulter).

### 4.6. Cell Sorting

BALF samples were filtered through a 70-μm cell strainer and centrifuged at 400× *g* for 10 min. The samples were stained with a combination of Brilliant Violet 421-conjugated anti-CD45 (30-F11, BioLegend), APC-conjugated anti-Ly6G (1A8, BioLegend), PE/Cy7-conjugated anti-CD11b (M1/70, BioLegend), and FITC-conjugated anti-SiglecF (S17007L, BioLegend). Immune cells were sorted using a Cell Sorter SH800S (SONY, Tokyo, Japan).

### 4.7. Quantification of Cytokines and Chemokines

The concentrations of TNF-α, IL-1β, IL-6, and CCL2 were quantified by flow cytometry using a BD Cytometric Bead Array (CBA) Mouse Cytokine Kit (BD Biosciences) according to the manufacturer’s protocol. CXCL2 levels were determined using an enzyme-linked immunosorbent assay (ELISA) kit (R&D Systems, Minneapolis, MN, USA). The levels of total proteins in the BALF were measured using a Pierce BCA assay (Thermo Fisher Scientific). The concentration of CGRP in the BALF was determined using an ELISA kit (Bertin Bioreagent, Montigny le Bretonneux, France).

### 4.8. AM Culture

AMs were isolated from BALF supernatants. Flow cytometry analysis revealed that >95% of cells in the BALF supernatants consisted of AMs [44]. The cell pellets were resuspended in Roswell Park Memorial Institute (RPMI) 1640 medium (Gibco, Thermo Fisher Scientific) containing 10% fetal bovine serum. The cells were adhered by incubation for 3 h (37 °C, 5% CO_2_ humidified atmosphere) in 12-well culture plates (5 × 10^5^ cells/well), and non-adherent cells were removed by repeated washing with fresh culture medium in the presence or absence of rat CGRP (100 nM; Peptide Institute) for 24 h, harvested, homogenized in RNAiso Plus (Takara Bio, Shiga, Japan), and mRNA levels measured by RT-qPCR. Supernatants from cultured AMs were collected to determine cytokine levels using a CBA kit (BD Biosciences), and data were analyzed using BD Biosciences FCAP software (v3.0).

### 4.9. BM-Derived Macrophage Culture

BM-derived macrophages were generated from BM cells obtained by flushing the femurs and tibias of 8-week-old male mice. BM cells were cultured in 6-well plates (1.0 × 10^6^ cells/well) and maintained in RPMI 1640 medium (Gibco) containing 10% fetal calf serum and 20 ng/mL macrophage colony-stimulating factor (BioLegend). On day 7, the BM-derived macrophages were stimulated with LPS (100 ng/mL; Sigma-Aldrich) in the presence or absence of rat CGRP (100 nM; Peptide Institute) for 3 h. The cultured BM-derived macrophages were harvested and homogenized in RNAiso Plus (Takara Bio), and mRNA levels were measured by RT-qPCR. Supernatants from cultured BM-derived macrophages were collected to determine cytokine levels using a CBA kit (BD Biosciences), and data were analyzed using BD Biosciences FCAP software (v3.0).

### 4.10. Adoptive Transfer of AMs

The adoptive transfer of isolated AMs (3 × 10^5^ cells in 50 μL PBS) was performed by i.t. delivery into mice anesthetized with a medetomidine-midazolam-butorphanol tartrate mixture at 24 h before LPS treatment. At 72 h after LPS administration, BALF and lung tissue samples were collected for analysis.

### 4.11. Dissection of Dorsal Root Ganglion (DRG)

Under anesthesia with the midazolam-medetomidine-butorphanol mixture, the DRGs of the Th4-12 vertebrae were isolated and collected [45]. Samples were prepared for quantitative PCR or immunohistochemical analyses.

### 4.12. Histology and Immunohistochemical Analysis

In separate experiments, the lungs were i.t. pressure-infused at 20 cm H_2_O with 4% paraformaldehyde-phosphate buffer solution and finally isolated. Lung tissue samples were immediately fixed with 4% paraformaldehyde in PBS at 4 °C for 24 h. Paraffin-embedded tissues were cut into 3.5-μm-thick sections, stained with hematoxylin and eosin (H and E), and imaged using a fluorescence microscope (Biozero BZ-700 Series; KEYENCE, Osaka, Japan).

The degree of tissue injury was determined according to the following lung injury scoring system [44,46]: 1, intrapulmonary hemorrhage; 2, presence or absence of congestion in pulmonary capillary; 3, presence or absence of neutrophil infiltration in lung interstitium; and 4, formation of a hyaline membrane (thickened alveolar wall). Five high-power visual fields (200× magnification) were randomly selected for each section. The lung injury score standard was as follows: 0, normal; 1, extremely mild damage (<25% of visual field); 2, mild damage (25–50% of visual field); 3, moderate damage (50–75% of visual field); and 4, severe damage (>75% of visual field).

DRG samples were fixed immediately with 4% paraformaldehyde in PBS at 4 °C for 24 h. The fixed tissues were embedded in paraffin, and 3.5-µm-thick sections were collected for immunostaining. The sections were blocked with a serum-free protein block (Dako, Glostrup, Denmark) and incubated overnight at 4 °C with anti-mouse CGRP antibody (rabbit polyclonal antibody, 1:200; Sigma-Aldrich). After washing in PBS, the sections were incubated using a Universal DAKO LSAB HRP system (Dako) with DAB and Mayer’s hematoxylin solution.

### 4.13. Immunofluorescence

Excised lung tissue samples perfused with 4% paraformaldehyde were fixed for 24 h, incubated in 30% sucrose and 0.1 M phosphate buffer at 4 °C for 72 h, embedded in OCT compound (Sakura Finetek, Torrance, CA, USA), cryosectioned to 13-µm-thick sections, and mounted on Superfrost slides (Thermo Fisher Scientific). The sections were incubated with the serum-free protein block (Dako) at room temperature for 1 h to block non-specific binding. The sections were then incubated overnight at 4 °C with the following primary antibodies: anti-mouse CD68 (rat monoclonal antibody, 1:100; Bio-Rad Laboratories, Hercules, CA, USA), anti-mouse myeloperoxidase monoclonal antibody (goat monoclonal antibody, 1:100; R&D Systems), anti-mouse IL-6 (rabbit polyclonal antibody, 1:100; Abcam, Cambridge, UK), anti-mouse aquaporin 5 (AQP5) (rabbit polyclonal antibody, 1:100; Abcam), anti-mouse prosurfactant protein C (SFC) (rabbit polyclonal antibody, 1:100; Abcam), anti-mouse MPO (goat polyclonal antibody, 1:100; R&D Systems), or anti-mouse RAMP1 (rabbit polyclonal antibody, 1:100; Alomone Labs, Jerusalem, Israel). Immunostaining with anti-mouse CGRP (rabbit polyclonal antibody, 1:100; Sigma-Aldrich) was performed on 80-µm-thick sections. After washing with PBS, the sections were incubated with the following secondary antibodies at 4 °C overnight: Alexa Fluor 488-conjugated donkey anti-rabbit IgG, Alexa Fluor 488-conjugated donkey anti-goat IgG, Alexa Fluor 594-conjugated donkey anti-rabbit IgG, Alexa Fluor 594-conjugated donkey anti-goat IgG, and Alexa Fluor 647-conjugated donkey anti-rabbit IgG (Molecular Probes, Eugene, OR, USA). Nuclei were detected with DAPI. All images were obtained using the fluorescence microscope (Biozero BZ-700 Series, KEYENCE).

### 4.14. Quantitative Real-Time Reverse Transcription-PCR Analysis

Total RNA was extracted from the lung tissues and homogenized using RNAiso Plus (Takara Bio). Single-stranded cDNA was generated from 1 μg of total RNA by reverse transcription using the ReverTra Ace qPCR RT Kit (Toyobo, Osaka, Japan). Quantitative PCR was performed using TB Green Premix Ex Taq II (Tli RNase H Plus; Takara Bio). PCR was performed in 20 μL reaction volumes and subjected to 40 cycles of amplification in a thermal cycler. The gene-specific primers used in these experiments are listed in Appendix A. Data were normalized to glyceraldehyde-3-phosphate dehydrogenase (*Gapdh*) mRNA levels in the same sample.

### 4.15. Statistical Analysis

All results are presented as the mean ± standard deviation (SD). All statistical analyses were performed using GraphPad Prism v8 (GraphPad Software, La Jolla, CA, USA). An unpaired two-tailed Student’s *t*-test was used to compare data between two groups. A one-way analysis of variance followed by Tukey’s post hoc test was used to assess differences between multiple groups. Survival rates were compared by the Kaplan–Meier survival analysis and log-rank tests. Statistical significance was set at *p* < 0.05.

## 5. Conclusions

This study showed that inhibition of RAMP1 signaling enhanced AM-mediated cytokine overproduction and neutrophil accumulation and suppressed appropriate inflammatory responses by MDMs following LPS stimulation, thereby inducing ALI. Targeting of RAMP1 signaling represents a promising therapeutic approach in the clinical management of ALI/ARDS.

## Figures and Tables

**Figure 1 ijms-25-10107-f001:**
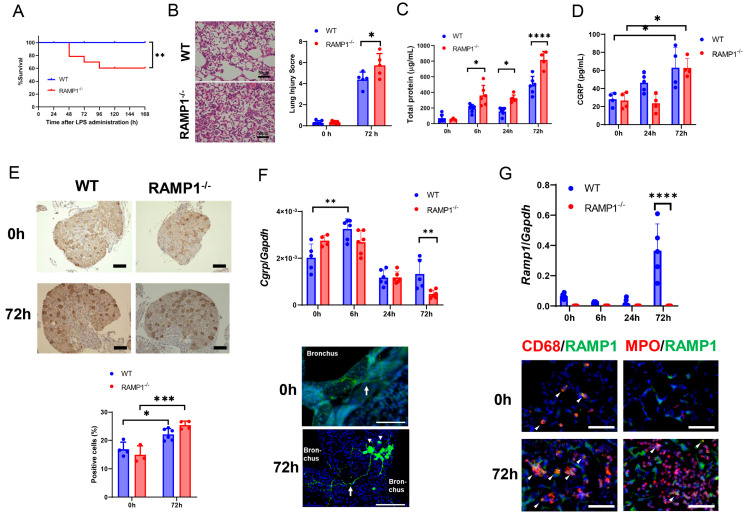
RAMP1 deficiency aggravates LPS-induced acute lung injury. (**A**) Survival rates of LPS-treated WT (*n* = 20) and RAMP1^−/−^ (*n* = 20) mice. Log-rank (Mantel-Cox) test. ** *p* < 0.01. (**B**) Representative photographs of H and E-stained lung sections from WT and RAMP1^−/−^ mice treated with LPS for 72 h. Scale bars: 50 μm. Lung injury score at 0 h and 72 h after LPS administration in WT and RAMP1^−/−^ mice. Lung injury score was assessed based on the presence of intrapulmonary hemorrhage, pulmonary capillary congestion, neutrophil infiltration in lung interstitium, and hyaline membrane formation. For detailed methodology, see Section 4.12 in the Methods. Data are expressed as the mean ± SD. * *p* < 0.05. (**C**,**D**) Concentration of total protein (**C**) and CGRP (**D**) in the BALF of WT and RAMP1^−/−^ mice after LPS treatment. Data are expressed as the mean ± SD. * *p* < 0.05 and **** *p* < 0.0001. (**E**) Representative photographs of immunostaining with CGRP in thoracic dorsal root ganglions (DRGs). Scale bars: 100 μm. The percentage of CGRP^+^ cells in DRGs is shown. Data are expressed as the mean ± SD. * *p* < 0.05, *** *p* < 0.001. (**F**) mRNA levels of *Cgrp* in lung tissues from WT and RAMP1^−/−^ mice. Data are expressed as the mean ± SD. ** *p* < 0.01. Representative photographs of immunostaining with CGRP (green) in lung tissue. Arrows indicate CGRP^+^ nerves. Arrowheads indicate PNECs. Scale bars: 50 μm. (**G**) mRNA levels of *Ramp1* in lung tissues from WT and RAMP1^−/−^ mice. Data are expressed as the mean ± SD. **** *p* < 0.0001. Immunofluorescence staining for RAMP1 (green) and CD68 (red) or MPO (red) in lung tissues from WT and RAMP1^−/−^ mice at 0 h and 72 h after LPS administration. Arrow heads indicate double-stained cells. Cell nuclei were stained with DAPI (blue). Scale bars: 50 μm.

**Figure 2 ijms-25-10107-f002:**
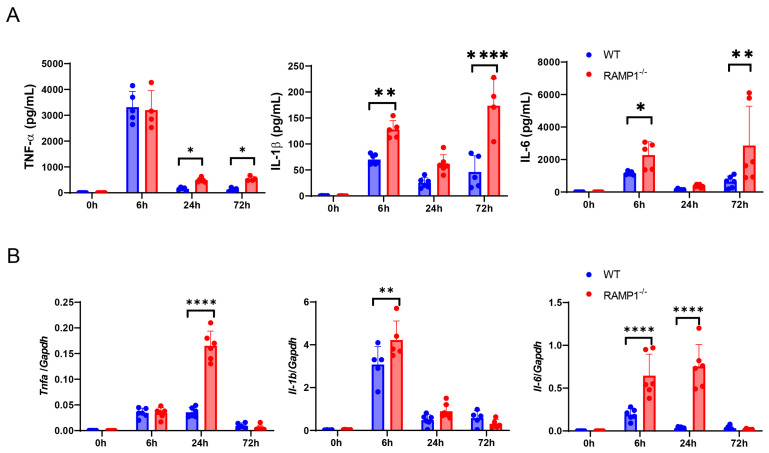
RAMP1 deficiency enhanced the production of pro-inflammatory cytokines. (**A**) Concentrations of the pro-inflammatory cytokines TNFα, IL-1β, and IL-6 in WT and RAMP1^−/−^ mice after LPS treatment. Data are expressed as the mean ± SD. * *p* < 0.05, ** *p* < 0.01, **** *p* < 0.0001. (**B**) mRNA levels of the pro-inflammatory cytokines *Tnfa*, *Il-1b*, and *Il-6* in WT and RAMP1^−/−^ mice after LPS treatment. Data are expressed as the mean ± SD. ** *p* < 0.01, **** *p* < 0.0001.

**Figure 3 ijms-25-10107-f003:**
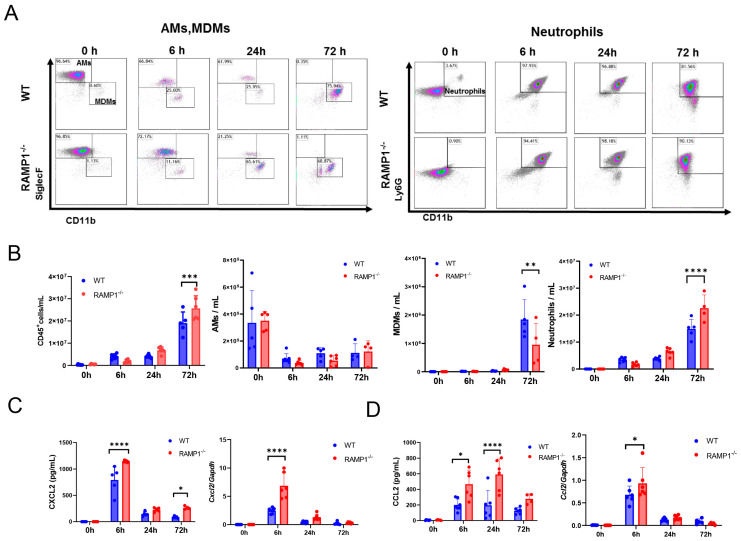
Changes in inflammatory cells during LPS-induced acute lung injury. (**A**) Representative dot plots for alveolar macrophages, recruited macrophages, and neutrophils in WT and RAMP1^−/−^ mice after LPS treatment. (**B**) Changes in the number of CD45^+^ cells, AMs (Ly6G^−^/SiglecF^high^/CD11b^low^ cells), MDMs (Ly6G^–^/SiglecF^low^/CD11b^high^ cells), and neutrophils (Ly6G^+^/CD11b^+^ cells) in the BALF of WT and RAMP1^−/−^ mice after LPS treatment. Data are expressed as the mean ± SD. ** *p* < 0.01, *** *p* < 0.001, **** *p* < 0.0001. (**C**) CXCL2 protein and gene expression in WT and RAMP1^−/−^ mice after LPS treatment. Data are expressed as the mean ± SD. **** *p* < 0.0001. (**D**) CCL2 protein and gene expression in WT and RAMP1^−/−^ mice after LPS treatment. Data are expressed as the mean ± SD. * *p* < 0.05, **** *p* < 0.0001.

**Figure 4 ijms-25-10107-f004:**
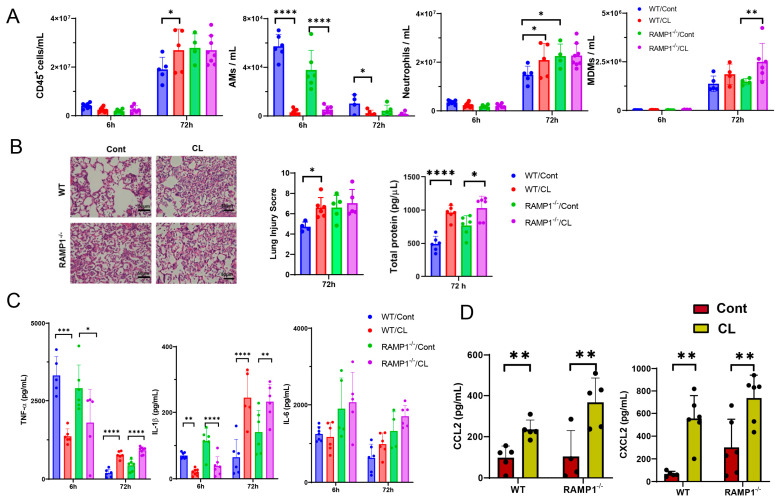
Inhibition of AM activity increased inflammatory cell levels, lung injury, and pro-inflammatory factor levels during LPS-induced acute lung injury. (**A**) Number of CD45^+^ cells, AMs, MDMs, and neutrophils at 6 h and 72 h after LPS administration in WT and RAMP1^−/−^ mice treated with clodronate liposomes (CL) or control liposomes (Cont). Data are expressed as the mean ± SD. * *p* < 0.05, ** *p* < 0.01, **** *p* < 0.0001. (**B**) Representative images of H and E-stained lung sections at 72 h in WT and RAMP1^−/−^ mice treated with CL or Cont. Scale bars: 50 μm. Lung injury score and total protein levels in BALF at 72 h after LPS administration in WT and RAMP1^−/−^ mice treated with CL or Cont are shown. Data are expressed as the mean ± SD. * *p* < 0.05, **** *p* < 0.0001. (**C**) Levels of the pro-inflammatory cytokines TNF-α, IL-1β, and IL-6 at 6 h and 72 h in the BALF of WT and RAMP1^−/−^ mice treated with CL or Cont. Data are expressed as the mean ± SD. * *p* < 0.05, ** *p* < 0.01, *** *p* < 0.001, **** *p* < 0.0001. (**D**) Levels of the chemokines CXCL2 and CCL2 at 72 h in the BALF of WT and RAMP1^−/−^ mice treated with CL or Cont. Data are expressed as the mean ± SD. ** *p* < 0.01.

**Figure 5 ijms-25-10107-f005:**
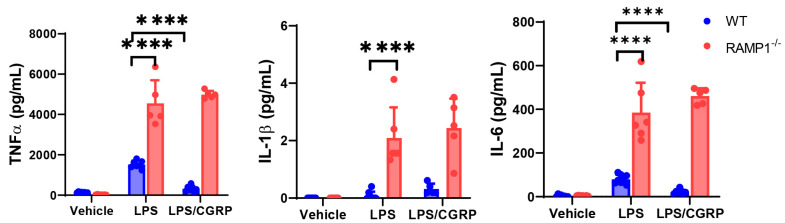
Expression of pro-inflammatory cytokines in cultured AMs from WT and RAMP1^−/−^ mice. Levels of TNF-α, IL-1β, and IL-6 in supernatants of cultured AMs stimulated with LPS in the presence or absence of CGRP. AMs were isolated from BALF samples. Data are expressed as the mean ± SD. **** *p* < 0.0001.

**Figure 6 ijms-25-10107-f006:**
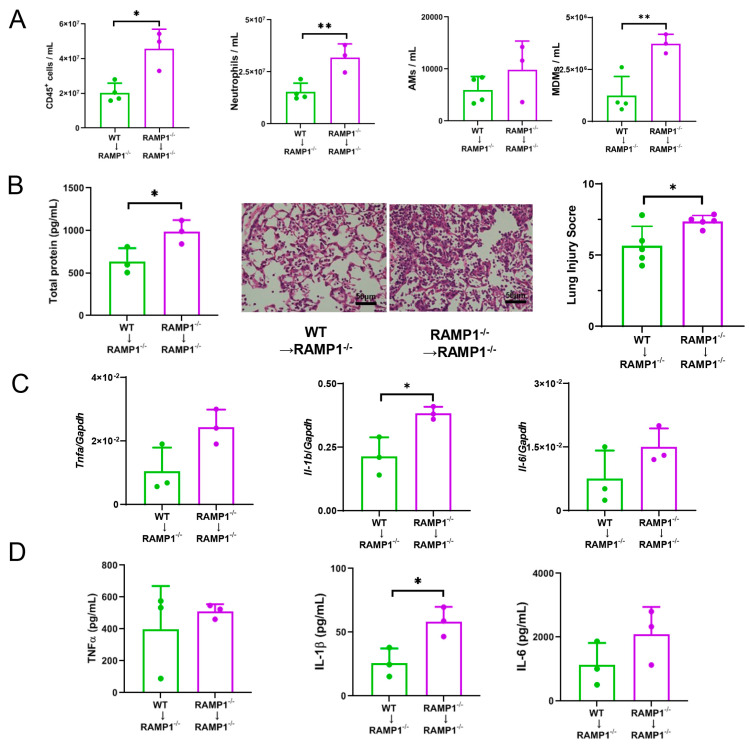
Effects of adoptive transfer of AMs on LPS-induced acute lung injury. (**A**) Number of CD45^+^-cells, neutrophils, AMs, and MDMs after adoptive transfer. AMs isolated from WT and RAMP1^−/−^ mice were transferred to RAMP1^−/−^ mice at 24 h before LPS treatment, and flow cytometry analysis was performed at 72 h after LPS treatment. Data are expressed as the mean ± SD. * *p* < 0.05, ** *p* < 0.01. (**B**) Levels of total protein in the BALF and lung injury score after adoptive transfer. Data are expressed as the mean ± SD. * *p* < 0.05. Representative images of H and E-stained lung sections from RAMP1^−/−^ mice transplanted with AMs isolated from WT or RAMP1^−/−^ mice are shown. Scale bars: 50 μm. (**C**) mRNA expression of *Tnfa*, *Il-1b*, and *Il-6* in lung tissues after adoptive transfer. Data are expressed as the mean ± SD. * *p* < 0.05. (**D**) Levels of TNF-α, IL-1β, and IL-6 in the BALF after adoptive transfer. Data are expressed as the mean ± SD. * *p* < 0.05.

**Figure 7 ijms-25-10107-f007:**
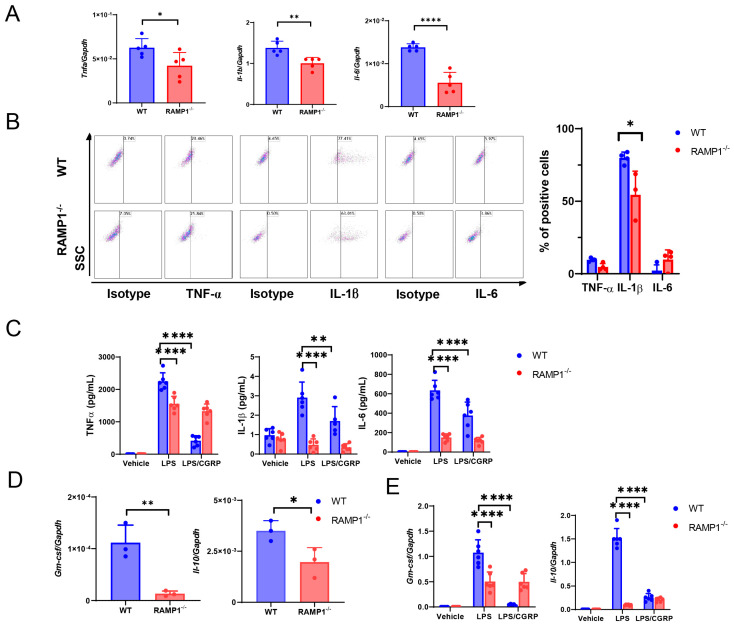
Pro-inflammatory and pro-repairing mediators in MDMs after LPS treatment. (**A**) Gene levels of the pro-inflammatory cytokines *Tnfa*, *Il-1b*, and *Il-6* in MDMs from WT and RAMP1^−/−^ mice at 72 h after LPS treatment. Data are expressed as the mean ± SD. * *p* < 0.05, ** *p* < 0.01, and **** *p* < 0.001. (**B**) Percentage of MDMs positive for TNF-α, IL-1β, and IL-6 in WT and RAMP1^−/−^ mice at 72 h after LPS treatment. Data are expressed as the mean ± SD. * *p* < 0.05. (**C**) Pro-inflammatory cytokines, including TNF-α, IL-1β, and IL-6, in supernatants isolated from BM-derived macrophages from WT and RAMP1^−/−^ mice stimulated with LPS in the presence or absence of CGRP. Data are expressed as the mean ± SD. ** *p* < 0.01, **** *p* < 0.0001. (**D**) Gene levels of *Gm-csf* and *Il-10* in MDMs from WT and RAMP1^−/−^ mice at 72 h after LPS treatment. Data are expressed as the mean ± SD. * *p* < 0.05, ** *p* < 0.01. (**E**) Gene levels of *Gm-csf* and *Il-10* in BM-derived macrophages isolated from WT and RAMP1^−/−^ mice stimulated with LPS in the presence or absence of CGRP. Data are expressed as the mean ± SD. **** *p* < 0.0001.

**Figure 8 ijms-25-10107-f008:**
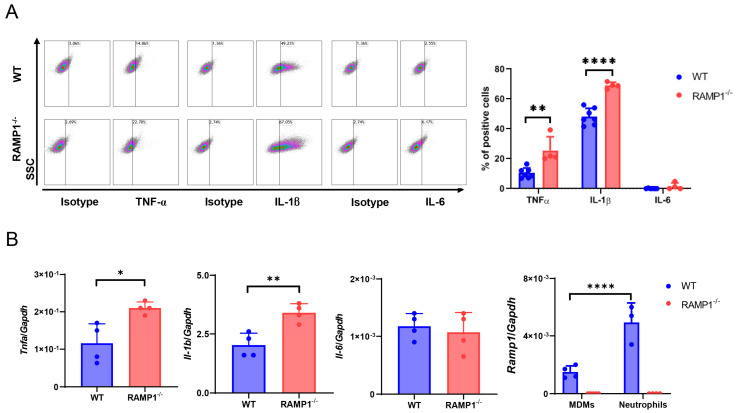
Inflammatory cytokines are released from accumulated neutrophils during LPS-induced acute lung injury. (**A**) Percentage of TNF-α^+^, IL-1β^+^, and IL-6^+^ neutrophils in WT and RAMP1^−/−^ mice at 72 h after LPS treatment. Data are expressed as the mean ± SD. ** *p* < 0.01 and **** *p* < 0.0001. (**B**) mRNA levels of the pro-inflammatory cytokines *Tnfa*, *Il-1b*, and *Il-6* and *Ramp1* in neutrophils and MDMs from WT and RAMP1^−/−^ mice at 72 h after LPS treatment. Data are expressed as the mean ± SD. * *p* < 0.05, ** *p* < 0.01, and **** *p* < 0.0001.

**Figure 9 ijms-25-10107-f009:**
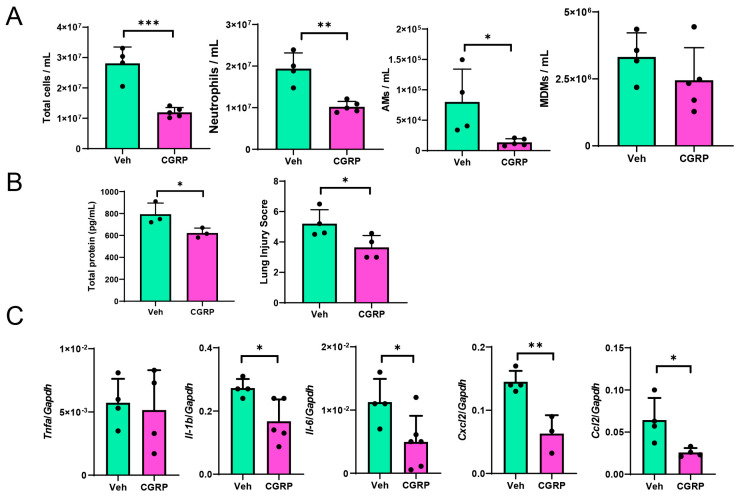
Effects of CGRP treatment on LPS-induced ALI. (**A**) Numbers of CD45^+^ cells, neutrophils, AMs, and MDMs at 72 h in WT mice treated with CGRP or Veh (vehicle). Data are expressed as the mean ± SD. * *p* < 0.05, ** *p* < 0.01, *** *p* < 0.001. (**B**) Total protein levels in BALF and lung injury score at 72 h in WT mice treated with CGRP or Veh (vehicle). Data are expressed as the mean ± SD. * *p* < 0.05. (**C**) mRNA levels of *Tnfa*, *Il-1b*, *Il-6*, *Cxcl2*, and *Ccl2* in lung tissues at 72 h after LPS administration in WT mice treated with CGRP or Veh (vehicle). Data are expressed as the mean ± SD. * *p* < 0.05, ** *p* < 0.01.

## Data Availability

The data supporting the findings of this study are openly available within the article or from the corresponding author upon request.

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
