# Peer review of "RAMP1 Signaling Mitigates Acute Lung Injury by Distinctively Regulating Alveolar and Monocyte-Derived Macrophages"

_ijms, 2024, doi:10.3390/ijms251810107_

Round 1

Reviewer 1 Report

Comments and Suggestions for Authors

It is reported that resident aveolar, but not monocyte-derived, macrophages play an obligatory inhibitory role in a mouse model of acute respiratory distress syndrome. This inhibitory effect is regulated by sensory nerve-produced calcitonin gene-related peptide and receptor activity-modifying protein (RAMP) 1. By pharmacological inhibition, genetic knockout mice, and adoptive transfer, the authors elegantly demonstrated that RAMP1 signaling mitigates LPS-induced ALI by inactivating AMs and promoting inflammatory and repair activities of MDMs.

Figure 3A, after LPS stimulation, alveolar macrophage significantly reduced. Do the authors have any explanation for it? What happens to these macrophages? Where are they going?

Figure 9, I did not find Figure 9c in the legend.

Author Response

It is reported that resident aveolar, but not monocyte-derived, macrophages play an obligatory inhibitory role in a mouse model of acute respiratory distress syndrome. This inhibitory effect is regulated by sensory nerve-produced calcitonin gene-related peptide and receptor activity-modifying protein (RAMP) 1. By pharmacological inhibition, genetic knockout mice, and adoptive transfer, the authors elegantly demonstrated that RAMP1 signaling mitigates LPS-induced ALI by inactivating AMs and promoting inflammatory and repair activities of MDMs.

We are grateful for the referee's thorough evaluation and constructive feedback regarding our manuscript.

Figure 3A, after LPS stimulation, alveolar macrophage significantly reduced. Do the authors have any explanation for it? What happens to these macrophages? Where are they going?

Thank you for your comments. In the text, we described the behavior of AMs following LPS administration (Lines 439-449).

During the initial phase of LPS-induced ALI, AM populations exhibit a significant reduction, which is consistent with previous findings [34]. Upon activation by LPS stimulation, AMs undergo cell death, resulting in decreased numbers of AMs. As dying cells release pro-inflammatory mediators, AM death plays a crucial role in the progression of inflammation during LPS-induced ALI. The reduction in AM populations may also be attributed to their detachment from the epithelial cells. In addition to apoptosis and necrosis, several modes of programmed cell death, including pyroptosis and ferroptosis, have been proposed; however, the precise mechanisms of AM death remain unclear [35] The repopulation of AMs occurs during the recovery phase of LPS-induced ALI, presumably through their self-renewal and local proliferation [20, 34,35].

Figure 9, I did not find Figure 9c in the legend.

We have corrected this (Line 378).

Reviewer 2 Report

Comments and Suggestions for Authors

Dear Authors,

Thank you very much for your well-written manuscript, presenting an interesting issue, which is the regulation of local signaling in LPS-induced ARDS. Please pay attention to the following comments and questions, pertaining to your manuscript:

1.      Line 76. Please provide the whole term for the abbreviation ALI (acute lung injury) or cite this abbreviation earlier in your text.

2.      Introduction. COVID-19 ARDS is a viral-induced lung injury, while LPS, which was used in your study, is a bacterial product. Can an LPS-induced acute lung injury represent the immune system activation in the lung of a viral-induced ARDS? Please comment.

3.      Lines 82-83. Please define that the mortality of your experimental mice was due to the acute lung injury and the subsequent severe acute respiratory failure.

4.      Were there any therapeutic measures (mechanical ventilation) to stabilize the animals over the maximum of the 96 hours until their death?

5.      Line 87. Please provide the whole term for the abbreviation BALF, it has been provided later in your text (line 134).

6.      Line 115. Please describe what the lung injury score is.

7.      Line 293. Please keep the same nomenclature of the abbreviation GM-CSF throughout your text. Right before (line 292) you used for example: Gm-csf.

8.      Line 302. Please correct as such: pro-repairing mediators.

Best Regards

Comments on the Quality of English Language

minor

Author Response

Dear Authors,

Thank you very much for your well-written manuscript, presenting an interesting issue, which is the regulation of local signaling in LPS-induced ARDS. Please pay attention to the following comments and questions, pertaining to your manuscript:

We are grateful for the referee's thorough evaluation and constructive feedback regarding our manuscript.

1. Line 76. Please provide the whole term for the abbreviation ALI (acute lung injury) or cite this abbreviation earlier in your text.

 We have corrected this (Lines 76-77).

2. COVID-19 ARDS is a viral-induced lung injury, while LPS, which was used in your study, is a bacterial product. Can an LPS-induced acute lung injury represent the immune system activation in the lung of a viral-induced ARDS? Please comment.

Thank you for your comment.

To investigate the role of RAMP1 signaling, we utilized a mouse model of ARDS induced by the administration of LPS, a component of the outer membrane of gram-negative bacteria, rather than a virus. Although the accumulation of various immune cell types in the lungs involved in the promotion of ARDS differs between LPS-induced ALI and virus-induced ARDS, both establish a cytokine storm, a characteristic feature of ARDS.

These were addressed in Lines 78-83.

3. Lines 82-83. Please define that the mortality of your experimental mice was due to the acute lung injury and the subsequent severe acute respiratory failure.

Mortality was defined by comprehensive pathological examinations during necropsy, which revealed macroscopically observable discoloration and hemorrhage of pulmonary tissue, as well as clinical manifestations including respiratory alterations (characterized by superficial respiration and elevated spontaneous respiratory rate), hypoactivity, weight reduction, and hypothermia.

This was addressed in the text (Lines 88-93).

4. Were there any therapeutic measures (mechanical ventilation) to stabilize the animals over the maximum of the 96 hours until their death?

To evaluate the natural progression of LPS-induced ALI, the mice were not subjected to therapeutic interventions (mechanical ventilation) throughout the course of this study.

This was addressed in the text (Lines 523-525).

5. Line 87. Please provide the whole term for the abbreviation BALF, it has been provided later in your text (line 134).

We have corrected this. Line 97.

6.  Line 115. Please describe what the lung injury score is.

Lung injury score was assessed based on the presence of intrapulmonary hemorrhage, pulmonary capillary congestion, neutrophil infiltration in lung interstitium, and hyaline membrane formation. For detailed methodology, see the section 4.12 in the Methods.

This was addressed in the text (Lines 125-127).

7. Line 293. Please keep the same nomenclature of the abbreviation GM-CSF throughout your text. Right before (line 292) you used for example: Gm-csf.

We have corrected this (Line 302).

8. Line 302. Please correct as such: pro-repairing mediators.

We have corrected this. Line 311.
